# A Kinect-Based Interactive System for Home-Assisted Active Aging

**DOI:** 10.3390/s21020417

**Published:** 2021-01-08

**Authors:** Gabriel Fuertes Muñoz, Ramón Alberto Mollineda Cardenas, Filiberto Pla

**Affiliations:** 1Edison Desarrollos, S.L., 44002 Teruel, Spain; 2Institute of New Imaging Technologies, Universitat Jaume I, 12071 Castellón, Spain; ramon.mollineda@lsi.uji.es (R.A.M.C.); filiberto.pla@lsi.uji.es (F.P.)

**Keywords:** Kinect-based interactive system, active aging, home-assisted healthcare, gamified environment

## Abstract

Virtually every country in the world is facing an unprecedented challenge: society is aging. Assistive technologies are expected to play a key role in promoting healthy lifestyles in the elderly. This paper presents a Kinect-based interactive system for home-assisted healthy aging, which guides, supervises, and corrects older users when they perform scheduled physical exercises. Interactions take place in gamified environments with augmented reality. Many graphical user interface elements and workflows have been designed considering the sensory, physical and technological shortcomings of the elderly, adapting accordingly the interaction methods, graphics, exercises, tolerance margins, physical goals, and scoring criteria. Experiments involved 57 participants aged between 65 and 80 who performed the same physical routine six times during 15 days. After each session, participants completed a usability survey. Results provided significant evidence that support (1) the effectiveness of the system in assisting older users of different age ranges, (2) the accuracy of the system in measuring progress in physical achievement of the elderly, and (3) a progressive acceptance of the system as it was used. As a main conclusion, the experiments verified that despite their poor technological skills, older people can adapt positively to the use of an interactive assistance tool for active aging if they experience clear benefits.

## 1. Introduction

The world’s population is experiencing rapid aging, which poses a challenge to nearly all areas of society: to meet a growing demand for specific goods and services adapted to the needs of the elderly. A striking example is the market of assistive technologies both for telecare and for promoting active aging, which is expected to grow dramatically in the coming decades [1]. Apart from the benefits in the quality of life, these solutions should entail a significant reduction in healthcare expenses [2].

Recently, a number of Information and Communication Technologies (ICT) have emerged aiming at providing innovative and effective ways to help seniors in their daily life [3]. In the meantime, studies have been conducted to bring assistive ICT closer to older people and to investigate their attitudes toward technology [4]. However, little work has been done to objectively assess the benefits of introducing technologies in the everyday life of the elderly.

Assistive ICT can be roughly classified into two categories: tools that focus on passive care and tools that promote active care. On the one hand, passive assistive technologies would comprise those tools designed to sustain everyday life [5]. Illustrative examples are full-fledged e-health technologies, which allow remote access to healthcare services for monitoring chronic diseases or for assisting in case of unexpected events [6]. A real-time remote monitoring of a person’s health is proposed in [7], ensuring early intervention in case of a sudden worsening of the health condition. Another example can be found in [8], where an activity tracking system assists Alzheimer’s disease patients to live independently. The system monitors a patient’s motion while they perform their daily activities and provides urgent assistance in case of need. Patients are expected to gain self-confidence, while caregivers are released from some of their burdens. On the other hand, active care would cover technologies for encouraging an active way of life by incorporating physical activity into the everyday routines of older people [9]. A recent study shows the potential of a virtual exercise class using gaming technology and avatars to improve physical capabilities and social connections among the elderly [10].

The benefits of physical therapy in adults aged 70 years and older have been well established in [7]. A study on frail nonagenarians assessed the effects of multicomponent training on muscle power output, muscle mass, muscle tissue attenuation, risk of falls, and functional outcomes [11]. Participants showed a significantly improved TUG (Timed Up and Go) with single and dual tasks, reflecting a better physical response. Similarly, the effects of intensive exercise training in community-dwelling older adults were examined in [12]. Results show that high-intensity activity improves measures of physical function more than low-intensity exercise routines [13]. Most of these initiatives were deployed through digital games [14], where about 90% of patients reported that using UERG games (Upper Extremity Rehabilitation Gardening game) increased their motivation [15]. Other studies offering game-based rehabilitation exercises [16,17,18] have also shown an increase in patient motivation and adherence to treatment and an objective progress in physical condition. A recent study [19] verified that pleasure and enjoyment play a decisive role in motivating older people. These efforts proved both the value of introducing physical therapy into a comprehensive care for the elderly as well as the acceptance of technology by older users when they perceive that these solutions bring verifiable benefits and elicit positive feelings.

Microsoft Kinect has also brought a revolution in technologies supporting physical therapy [20]. The integration of Kinect, gaming, and virtual reality for physical rehabilitation of patients with brain injuries can be found in [21,22,23]. They focused on strengthening the motivation of patients to perform exercises and on training the brain to recover a lost ability by roughly asking the patient to approach a goal. However, these proposals do not involve real-time correction mechanisms nor do they objectively quantify the level of achievement of a physical goal. A Kinect-based system with an interactive augmented reality environment was successfully used for the rehabilitation of upper limbs in stroke patients [24]. Based on motion data obtained from measuring devices, the system monitors and assesses the rehabilitation progress. In a more recent effort [25], a system called KineActiv was designed to replace physiotherapists in supervising upper limb exercises, helping patients affected by some arm injury to achieve faster rehabilitation. Nevertheless, despite the rehabilitation potential of these tools, they are not sensitive to age-related functional limitations such as vision and hearing loss, declining motor skills, cognitive impairment, and poor technological culture [26].

### 1.1. Related Work

A comparison between Kinect Adventures games and conventional physiotherapy to improve postural control, gait, cardiorespiratory fitness, and cognition in the elderly was carried out in [27]. Other areas such as acceptability and adherence to treatment were also analyzed. Experiments were carried out with 23 older participants in each group, whose capacities were measured through conventional techniques before and after the treatment. The results of statistical tests found benefits after both types of intervention in practically all the areas, but there were no significant differences between both treatments. Another study on the potential of Kinect-based exergaming, compared to traditional exercises, in improving frailty status and physical performance in the prefrail and frail elderly is presented in [28]. The experiments involved 52 older people in two groups, who received aerobic, resistance, and balance training through Kinect-base exergaming and combined exercises, respectively. After 36 sessions over 12 weeks, the improvements in physical condition in the group using Kinect were greater than or equal to those in the group trained by conventional methods. In these two works, Kinect was used as an external motivational tool, without any system to supervise and control the execution of prescribed exercises, or to measure physical achievements.

The reliability of Kinect-based measurements on forward reach distance and velocity tasks was explored in [29]. The experiments were conducted on 442 older participants with a mean age of 73.3 years. Results showed good correlations between the Kinect-based outputs and measurements obtained by traditional techniques, showing that Kinect provides reliable and age-sensitive measurements: physical performance was significantly lower in individuals older than 75 years. They used the data streams provided by Kinect to calculate measures of physical performance in tasks related to gait and balance. However, the developed software lacked immersive user environments for motivational purposes.

The suitability of using hand gesture to provide home-assistance healthcare services for elderly patients was investigated in [30]. A Kinect-based real-time system for hand gesture recognition was implemented. Installed in front of the elderly patient, the system recognizes hand gestures and sends associated requests to care providers. Experiments showed promising results on the use of hand signals as a reliable and comfortable means of communication for older patients unable to convey their needs or feelings through words.

A recent study [31] introduces a Kinect-based system for calculating the shift angle of the gravity center of 80 older people during walking, with the aim of objectively estimating the fall risk probability in the elderly. The risk of falling is also the object of study of [32]. A gait measurement method is proposed to evaluate motor function in well-known dynamic gait tests using Kinect sensors. It looks for the best matching between a predefined model and the subject’s model during the task and estimates joint and angle trajectories. The results were compared to measurements from a three-dimensional motion capture system, proving the usefulness of the low-cost estimates provided by the proposed system for assessing clinical tasks. These systems are designed for use in specialized environments (e.g., clinical institutions) and therefore do not cover the scope of user environments aimed at older people.

A proposal similar to the one presented here is introduced in [33]. It is a Kinect-based prototype that implements an augmented reality exergame to motivate elderly people to perform physical exercise at home. The system includes several levels of difficulty that can be overcome by performing certain physical exercises. For example, the user should imitate flying by repeatedly lifting the arms to the shoulder height, parallel to the ground, with the elbow joint straight. Despite the fact of being the closest proposal to ours, there are some key differences: (1) although the work is interested in the user experience, it does not include any usability study; (2) this prototype was tested with only three participants aged between 43 and 62 years, which is a too small sample whose profile does not correspond to elderly; (3) although the system checks the correctness of the exercises, it does not appear to provide numerical measurements that allow performance to be compared between different sessions; (4) this prototype seems to implement a single environment, for 52 exercises and 34 exergames in our system. In short, this is an interesting work, but with much less functional scope, usability analysis, and experimental work than the one proposed here.

Another recently published system based on Kinect that promotes and controls the execution of physical exercises is described in [34]. The system implements a game-driven measurement function, which only recognizes the squat movement by comparing the shoulder position with a threshold when it moves up and down. It measures the optimal exercise time and includes a scoring feature. Experiments with 10 older people aged between 70 and 90 showed the usefulness of the system for active aging. Despite the validity of the proposal, some limitations can be identified. The squat control algorithm depends exclusively on the shoulder joints, so it does not check the knee angle or lower limbs joints. Experiments do not include a numerical evaluation or a usability study, and the latter are particularly important in the case of elderly users.

Table 1 summarizes a comparison between reviewed papers and the proposed work based on a common set of criteria that include system features and strategies for evaluating the system effectiveness. System features focus on usability technologies (e.g., augmented reality, gamification) and system capabilities to monitor, control, and measure movement in real time, while evaluation strategies consider the amount and profile of data, the nature of measurements, and the use of statistical tests to validate the significance of the results.

All things considered, none of the reviewed papers can be considered directly comparable with ours in terms of functional scope, control algorithms, and diversity of user environments and interaction modes.

### 1.2. Contributions

This paper presents a system inspired in KineActiv focused on the elderly that integrates age-sensitive interactive strategies in an augmented reality environment to assist older people in carrying out physical activities in a domestic environment. The main differences between the new system and KineActiv are summarized below:The goal of the new system is to help older people and prevent or mitigate the frailty state of the elderly, while the goal of KineActiv is the rehabilitation of a joint injury mainly in young adults. According to [35], frailty is the result of the cumulative decline in multiple physiological systems over a lifespan. This difference fosters distinct methods of controlling movement, evaluating achievements, interaction modalities, etc.The movement control mechanisms in the new system (e.g., methods, tolerance margins, achievement levels, etc.) have been adapted to facilitate the performance of physical exercises to older people. Physical exercise in old age reduces the loss of muscle strength and muscle mass, which are relevant symptoms of frailty [35]. Conversely, the control in KineActiv is strict, as the goal is to regain the full range of motion of an injured joint.The new system significantly reduces the need for explicit user interaction through more natural and intuitive interaction techniques such as gesture recognition and customizable time delays between screens. It is aimed at simplifying the use of the system by older users with poor technological skills. These interaction modes are not available in KineActiv.New features of augmented reality, gamification, and reward policies have been implemented in the new system, in order to strengthen motivation and user experience. In particular, the new system implements 34 AR-based gamified environments, which is a much larger amount than the 15 environments developed in KineActiv.The new system implements 52 physical exercises divided into 29 and 23 corresponding to the shoulder (upper limbs) and knee joints (lower limbs), respectively, in all cases adapted to the reduced body competence of the elderly. This amount is much higher than the 19 routines implemented in KineActiv, all of which are related to the shoulder (upper limbs).

To our knowledge, as depicted in the Table 1, no other work has reported a home-assisted tool that is able to automatically perform a fine control and measurement of limb movement in older people, while they performed guided physical exercises prescribed by a specialist. Although this is intended to replace the supervisory role of a human in home environments, a remote control of each user’s progress and their adherence to prescribed routines can be performed from a web-based application. This continuous feedback allows the routines to be dynamically adjusted to each user. Expected benefits for the elderly in the medium and long term are manifold: the objective improvement of health capacities should favor a greater autonomy, lower fall risks, as well as positive feelings such as joy, enthusiasm, and self-esteem.

In addition to the functional scope described, to the best of our knowledge, no previous work has carried out experiments aimed at showing both the acceptance and the effectiveness of a tool on a representative number of older users with balanced distributions between genders and age ranges. In this work, comprehensive experiments based on measurements from physical activities and responses from usability surveys were designed to statistically establish the extent to which these expectations are met. A total of 57 participants were recruited with 29 males and 28 females aged between 65 and 80. Results supported by two non-parametric tests proved that despite their age-related functional limitations, older people adapt positively to the use of multimedia assistance tools focused on active aging if they experience clear benefits.

All things considered, the main contributions of this paper can be summarized as follows:A Kinect-based system designed to assist older adults in incorporating physical activity into everyday routines through augmented reality environments, gamification, gesture-based user interfaces, and other interaction modalities adapted to their age-related functional decline.Experiments aimed at assessing both the effectiveness and the sensitivity to age of the system in measuring physical achievement of older people when performing prescribed exercises.Results from the usability surveys suggest a progressive acceptance of technological tools by older people that bring them tangible benefits to their health.

The rest of the manuscript is organized as follows. The next section introduces the technical and methodological foundations underlying the system. Section 3 presents the experimental set-up designed to validate the system validity, which is based on a usability survey, a number of physical exercise routines, and two statistical tests. Section 4 discusses the experimental results, while the main conclusions are presented in Section 5.

## 2. Active Aging System: Materials and Methods

This section describes the RGBD (Red, Green, Blue, Depth) sensor-based interactive system for guiding older users in performing prescribed physical activity within an active aging strategy. As discussed above, this system extends the functional scope, the control mechanisms, and the user interaction modalities of a previous tool (KineActiv) [25] to help older people mitigate the age-related motor frailty of the elderly. KineActiv was designed to replace physical therapists in rehabilitation sessions of upper limb injuries in young and middle-aged people who have good motor skills and technology acceptance behaviors. Figure 1 illustrates KineActiv’s main workflows, where a gamified user interface based on Augmented Reality (AR) is connected to a distributed system with a central server and a database. A web-based application is used by the physiotherapist to prescribe treatments and to monitor the user’s progress. The gamified environment is intended to make the rehabilitation process more friendly and enjoyable, using specific gamified contexts for each type of exercise.

The following subsections summarize the technical foundations of the Kinect device (Section 2.1), the functional scope of the proposed system (Section 2.2), as well as methods implemented to support AR-based user interfaces (Section 2.3), gamified environments (Section 2.4), gesture-based user interactions (Section 2.5), and real-time assessment of the validity of exercises (Section 2.6). Finally, Section 2.7 presents the most important tools used in system development.

### 2.1. Kinect as a RGBD Sensor: Technical Foundations

Kinect v2 is a low-cost motion sensing input device composed of RGB cameras and sensors that is able to map depth through time-of-flight computation at a spatial resolution of 512 × 424 pixels and at a depth working range from 0.4 to 4.5 m. It is supported by a Software Development Kit (SDK) that builds human skeleton models of up to six people present in the scene, tracks human motion, and recognizes gestures in real time, among other applications. All these capabilities turn Kinect into a natural user interface that does not require any physical interaction.

The SDK provides a number of data streams, the most popular being those comprising 2D color images, 3D depth images, and 3D skeletal frames. The latter includes the built-in skeletal model consisting of the 3D locations of 25 joints, as shown in Figure 2. As stated in [36], this model is the first step toward human motion recognition. Once the human skeleton is estimated, the motion should be interpreted, and appropriate feedback should be delivered to the user.

The free SDK paved the way for developing a wide range of applications, some of them aimed at supporting physical therapy and rehabilitation. They are mainly intended to ensure an automatic at home supervision of repetitive exercises prescribed by medical specialists, based on tracking marker-less body joints and segments. Figure 3 shows an example of motion tracking through the built-in skeleton model.

### 2.2. Functional Scope

With the aim of exploiting the potential of gamification and AR interfaces for improving motor skills in older adults, the following functional scope were identified:Natural sensing techniques: More natural and intuitive interaction techniques such as gesture or voice recognition, while avoiding elements difficult to manage by elderly people such as buttons to change the screens.Loose tolerance margins: Loose criteria for assessing movement correction in the course of an exercise execution to compensate for the limited motor skills of the elderly people.Gesture and scripting-based user interface: Natural and easy gestural inputs to interact with the system, either to start the game or to play with it. The interface can react by deploying a script-based scheme, in which the user simply has to follow step-by-step indications.AR environment: A pleasant AR environment that integrates the user as well as provides real-time feedback on the status of the current exercise.Rewarding scheme: Gamification offers a natural interactive environment to implement rewarding schemes to keep users’ attention and make the system more appealing. Thus, gamification can be of more importance for older users with poor technological skills.Improved graphic elements: Graphic elements, such as games and avatars, should contribute to make the system more understandable and enjoyable for older users.

The typical interaction flow for an older user to perform a prescribed exercise can be outlined as follows. To start a new session, the user must authenticate through username and password. This is the unique keyboard or mouse interaction during the session. As a future improvement, biometric identification through facial or voice recognition is planned. Once logged in, the system evaluates the user position in the scene and the body posture, and it provides visual feedback if any correction is needed. Then, the user navigates through multiple screens that provide instructions on the prescribed exercise. Customizable time delays between screens eliminate the need for explicit user interaction. After the last information screen, the system pauses until the user performs a gesture required to start the exercise: the hand with the palm facing the sensor moves from right to left or from left to right (Figure 4). This action is illustrated on the screen to keep the user engaged. Once the gesture is recognized, the exercise begins. The system deploys a customized gaming environment where goals are represented through AR visual elements. After being informed about objectives and repetitions, the user starts to perform the activity. The system checks whether each repetition meets the pre-set goal, considering the configurable loose tolerance margins, and it provides real-time visual feedback on the level of achievement. At the end of the routine, all series of measurements are stored in a central database located on a server. Then, physiotherapists can access statistics and charts that summarize the results of exercises and reschedule new sessions accordingly.

From the above description, four user interaction features can be elicited as those that play a major role in bringing the system closer to the elderly:AR-based user interfaceGamified environments with rewarding policiesGesture-based user interactionsReal-time assessment of the level of achievement based on loose tolerance margins.

The next subsections discuss the methods involved in the implementation of these features.

### 2.3. AR-Based User Interface

The graphical user interface includes a number of visual elements that were designed both to guide users during the exercises and to provoke in the user a feeling of integration in the environment. A very prominent example of the first objective is the use of avatars, which were designed as 3D animations to illustrate the correct way to perform exercises. They proved to be a dynamic, self-explanatory, and appealing resource for teaching users, particularly elderly ones. Figure 5 shows an example of a 3D avatar exemplifying a double leg squat.

To meet the second objective (feeling of integration), dynamic AR components were added to gamified environments. Figure 6 shows an example where AR elements represent the goal of a given exercise. In this case, the 3D coordinate of a rendered bird cage in the user scene indicates the position to which the user should raise their arm in a shoulder abduction exercise to meet the goal. Reaching the target position of the exercise is represented by putting the bird into the cage. Once the user has completed the exercise, the bird gets out of the cage and flies away.

Inserting AR elements into a scene requires contextual information from the scene and biometric data from the user. For instance, the computation of the 3D coordinates of the bird cage that represents the exercise goal (Figure 6) depends on the user position in the scene, the target angle, and the length of the user’s upper limb. More formally, given a particular exercise, the joints and muscles involved in its correct execution are retrieved from the database, together with some assessment rules that rely on the exercise goal and physical measurements between the joints of interest of the user.

Each exercise begins with a calibration step. Once the body is correctly positioned on the scene, the 3D coordinates of the joints of interest are obtained from the stream of 3D skeletal frames, and segments between these joints are modeled through vectors of biometric nature. As a way of example, a calibration related to shoulder exercises includes the locations of the shoulder, elbow, and wrist. Let *S*, *E*, and *W* represent these three joints, respectively. Then, vectors $\vec{SE}$ and $\vec{EW}$ will denote the arm and the forearm, respectively. A similar calibration example is performed for the knee exercises, where locations of the hip (H), knee (K), and ankle (A) are elicited. As in the case of the shoulder routines, the vectors $\vec{HK}$ and $\vec{KA}$ are estimated. Figure 7 illustrates both scenarios. The properties of these vectors can be considered user-specific and, therefore, invariant between different sessions. This customization process is conducted with the user placed in front of the camera in a relaxed posture in order to establish the initial resting state of each joint. This biometric data will be stored and retrieved at the beginning of the related exercise, and they will be used as a reference in the gamified environments.

Calibration results feed the method of estimating the 3D coordinates of the AR elements that represent the exercise goal (e.g., a target angle). These coordinates are the benchmark against which to measure the achievement of the exercise goal with a precision of 5 decimal places. Whether the goal is attained or not, the result of each repetition is measured (e.g., the observed angle) and stored in the database for future analysis. In addition to evaluating the final state of the exercise, the positions of the joints and the related vectors are updated 30 times per second to ensure that the complete execution of the exercise is correct. This process is based on some tolerance margins that are also customized for each patient. The role of these margins will be discussed later.

AR technology has also been used to identify the person who is receiving the system’s attention by attaching a visual mark on the user’s chest in the video stream, simulating a logo printed on the user’s shirt. This allows keeping multiple people in the scene, e.g., a physiotherapist and a user, with the system visually indicating who is being monitored. In this way, the system keeps the user informed that they are being recognized as the subject of interest. The mark, consisting of a circle that encloses the system logo, is placed on the coordinates associated with the joint called SPINE_MID (see Figure 2), which is located around the thoracic T5 vertebrae. The mark moves together with this joint in a way that keeps the user identified throughout the entire session. Figure 8 depicts the mark and the SPINE_MID joint in isolation.

The size of the mark was chosen in such a way as to allow a correct visualization, but without interfering with the rest of the elements of the graphical user interface or with the closest joints (SPINE_BASE and SPINE_SHOULDER).

### 2.4. Gamified Environment with Rewarding Policies

Each type of exercise is linked to a game. Games have been designed prioritizing simplicity, and keeping in mind the two objectives discussed above: (i) to guide the user and (ii) to maintain the user’s interest until the exercise is completed.

Each of the games chosen for the implementation of the exercises has followed the premise of having common elements based on the type of movement to be performed in order to have a common line and thus maintain consistency with respect to the user. For example, for concentric exercises in the upper joints, the same game is always used, which is adapted to the needs of the movement. The aim of using these criteria is to provide coherence for the 34 created environments and avoid monotony, making the interaction with the system more attractive and adapting all AR elements based on the suitability and diversity of movements.

Games also include a ludic and rewarding feature that consists in scoring the execution of exercises and showing the marks in rankings involving other users’ scores. Once the exercise is over, the resulting score and the updated ranking are displayed. The first part, as explained in the previous section, consists of comparing a target magnitude (e.g., an angle, a time frame, etc.) against an observed magnitude. The second part is more specific, as it depends on the nature of each exercise. As an example of evaluating the execution of an activity, a shoulder exercise called “scapular retraction” will be explained. Scapular retraction involves pulling the shoulder blades (scapulae) back toward the spine so that both scapulae are as close to each other as possible. Figure 9 and Figure 10 illustrate this exercise.

The control of the execution of this exercise is based on the SHOULDER_RIGHT, the SPINE_SHOULDER, and the SHOULDER_LEFT joints (see Figure 2). Let R, S, and L denote these three joints, respectively, and $\vec{RS}$, $\vec{LS}$, and $\vec{RL}$ denote the vectors between them. A top view of these joints resembles a triangle defined by the three vectors (see Figure 11 left). Note that unlike the previous examples, these vectors do not represent body structures and, therefore, they are expected to change during the execution of the exercise.

The assessment rule measures the lengths of the vectors in real time and evaluates certain heuristic conditions that validate the expected deformations when shoulders approach each other, both moving away from the sternum. These conditions can be summarized as follows:



$\left|{\vec{RS}}_{t}\right|$
≥
$\left|{\vec{RS}}_{t-k}\right|$

$\left|{\vec{LS}}_{t}\right|$
≥
$\left|{\vec{LS}}_{t-k}\right|$

$\left|{\vec{RL}}_{t}\right|$
≤
$\left|{\vec{RL}}_{t-k}\right|$
abs$\left(\left|{\vec{RS}}_{t}|-|{\vec{LS}}_{t}\right|\right)<\epsilon$where t, k, and ϵ denote the current time instant, a configurable time shift, and a parameter to control the symmetric execution of the exercise, respectively.


When all the conditions are simultaneously met from the beginning to the end of the exercise, it is considered correctly executed. The parameter *k* allows adjusting the precision to control the deformation of the vectors: small values lead to demanding rules, while larger values promote more permissive executions. When a rule is not met, the user is informed precisely about what part of the exercise (e.g., which shoulder) should be corrected. In this system, which is designed to assist the elderly, the values assigned to the parameters *k* and ϵ are aimed at adapting the rules to the motor skills of each user.

Figure 11 shows the gamification environment designed to encourage users to perform an isometric execution of the scapular retraction correctly. Isometric exercises involve a static contraction of a muscle or group of muscles without any noticeable movement in the angles of the affected joints. The goal is to hold muscle tension for a period of time.

This is an example of the exercises that refer to the trapezius muscle, which has several different exercises, all of which have the theme of pirates, instead of aliens, which was used, for instance, in concentric exercises. The game is about a pirate ship that attacks a tower on an island with a cannon. The firepower of the cannon will be proportional to the time the user holds muscle tension during the isometric scapular retraction. The user wins if the projected bullet hits the tower.

### 2.5. Gesture-Based User Interactions

Human gesture recognition aims to interpret the semantics of human gestures. A gesture can be defined as any movement of the face, hands, or other body parts that is intended to convey a message. Therefore, gesture recognition is a particular form of motion recognition. Nowadays, video-based human gesture recognition is gaining momentum as one of the more promising human–computer interfaces due to its simplicity and intuitiveness [38].

Taking advantage of the many opportunities that Kinect V2 offers to recognize human motion, a rule-based approach has been developed to recognize a simple and natural gesture that allows the user to interact with the system. In particular, a rule to recognize the movement of the hand from left to right and from right to left has been implemented, with the palm facing the sensor. Once the system recognizes this gesture, it automatically changes to the next screen. For example, this action would allow starting a game linked to the selected exercise.

This rule-based recognizer identifies both hand joints, HAND_LEFT and HAND_RIGHT, represents them through gray circles, and waits for one of the hands to make a movement that fits the expected pattern. The starting position of the hand in this motion pattern is set at half the user’s height on the *y*-axis, and away from the trunk on the *x*-axis (see Figure 12, left). When one of the hand joints is detected in a valid starting position, the color of the circle representing that hand changes to green to inform the user that the starting position has been validated. As long as the user moves the hand along the *x*-axis, keeping the *y*-coordinate approximately constant, the green circle accompanies the execution, indicating that it is still recognized as valid (see Figure 12, center). Finally, when the hand exceeds the *x*-coordinate of the SPINE_MID joint, the gesture is considered completed, which is an event that is represented by changing the color of the circle from green to light gray (see Figure 12, right). Once the gesture has been fully identified, the system changes the screen, and the exercise begins.

Recognizing human gestures through Kinect V2 is a hot topic, as it was extensively surveyed in [38]. After this first experience, we plan other gesture-based features in future versions.

### 2.6. Real-Time Assessment of the Level of Achievement based on Loose Tolerance Margins

The system includes several mechanisms to evaluate the real-time execution of exercises and the achievement of objectives. In addition to algorithms such as the one described in Section 2.4, to control the performance of scapular retraction, the validity of many exercises is assessed through tolerance margins, as reported in [25]. In that work, a tolerance margin was defined as a customized 3D region outside of which any activity was deemed invalid. These tolerance margins are usually considered together with postural margins, which are intended to avoid compensatory postural changes.

To adapt the system to the reduced body competence of the elderly, all the margins involved in the evaluation of the implemented exercises have been relaxed. For example, in older users, an abduction or flexion of the arm usually involves a compensatory torso tilt that affects the posture of the body. In this case, the torso tilt tolerance has been extended from 5 to 20 degrees tilt, which is about 15 degrees above the margin established for young adults without injury. In this way, older people can perform shoulder exercises more comfortably.

Another example is the adaptation of the margin established for the squat exercise. Figure 13 shows (in black) the correct position of the knees during a standard execution, where none of the knee joint is allowed to exceed the tip of the feet for it to be considered valid. However, this restriction has been relaxed for older people, allowing a deviation of up to 10 cm.

### 2.7. Implementation

Figure 14 shows the set of exercises implemented, which have been divided into shoulder and knee activities. Since the shoulder has more degrees of freedom than the knee, the number of exercises of the former is higher than that of the second. They were chosen according to suggestions made by the medical team, with the aim of addressing the most common locomotor problems in old age, and thus, of improving the quality of life in the elderly. An example illustrating how a user interacts with the system is shown in [39].

Despite the diversity of exercises, the execution of all of them follows a common flow diagram illustrated in Figure 15. By means of a gesture-based command, the user lets the system know that they are ready to perform the exercise, and the system begins to monitor the movement, particularly the joints and segments involved in the exercise. The system waits until the user’s position is recognized as the correct one to begin the first repetition of the exercise. When the user starts to execute this first repetition, the system monitors their movements and checks in real time that they meet the control rules defined for the exercise, which is usually based on tolerance margins. If an error is detected, the user is informed through visual feedback, so that they can try to correct it in the next repetition. This process continues until all scheduled repetitions are completed, after which the achievements are stored in a database.

The main software tools used for the design and implementation of the sensor management and interactive components were the following:Autodesk Maya: The new 3D avatars were built using the Maya 3D animation software. To better illustrate the exercise, avatars were designed to rotate 360º while performing the movement, thus making observation possible from all angles.Adobe Illustrator: The 2D graphic elements for gamification were totally redesigned with respect to the original system to follow a unified and consistent style. Adobe Illustrator was used for this purpose. To create the motion effects of the 2D graphic elements, all the necessary frames were designed and sequenced to compose the desired video animation.Unity engine: The gamified environment was implemented using the Unity engine, which also allowed us to exploit the full potential of the Kinect sensor.

## 3. Experimental Methodology

Experiments were intended to meet a dual purpose: to measure the degree of acceptance of the system by older people and to assess the effectiveness of the system in guiding the elderly in carrying out the exercises and measuring their physical achievement. Acceptability was judged through a usability survey of nine questions, while the system effectiveness, in terms of statistically verified progress of physical achievement over a series of sessions with goals of increasing complexity.

The key criterion to establish that expectations are met will be the presence of statistically significant improvements between measurements (survey responses, physical achievements) spread over a period of 15 days.

Each participant took part in six sessions over a 15-day stay in a center specialized in caring for the elderly, completing the same physical routine and the same survey in each session. Once the routine had been performed, participants were asked to fill out the survey in order to assess whether their perception of the system changed as they faced it every time.

The next sections present the demographic profile of participants (3.1), the fundamentals of the acceptability study (3.2), the physical exercises used as a benchmark to measure the effectiveness of the system (3.3), and the statistical tests that will establish the significance of results (3.4).

### 3.1. Participants

The study involved 57 participants aged between 65 and 80 years, of whom 29 were men and 28 were women. Figure 16 illustrates the distribution of men and women by age ranges. As observed, the numbers of men and women in each age range remain balanced.

People without known severe neurological or physical disorders were eligible, as the study focuses on routines for improving physical capacities in healthy older adults. Participants signed a written consent form, agreeing to get involved in the study provided that their personal data will remain confidential and secure according to present regulations.

### 3.2. The Usability Survey

The acceptability study was formulated in terms of a usability survey, which was designed considering both the well-known System Usability Scale (SUS) [40] and some popular categories: Satisfaction, Ease of Use, Happiness, Importance, and Usefulness. Due to the fact that the SUS model (see Table 2) suffers from redundancies (e.g., items 2 and 8) and contains both negative and positive questions (e.g., items 2 and 3), its direct use was discarded. Both factors could add unnecessary complexity to a questionnaire that was aimed at older users. The resulting survey is summarized in the Table 3, where each question is related to a category and to one or more SUS items.

For the sake of consistency regarding user understanding, all questions were framed in positive terms and scored from 1 to 5, with each score representing the user’s agreement with the related question. Table 3 suggests that the proposed questionnaire is more straightforward, consistent, and complete than a plain adaptation of the SUS model. As can be seen, it includes questions to judge the importance of gamified environments and the value of the system for the user, which are aspects that are not considered in the original SUS template. This design is expected to contribute significantly to collecting more reliable information from people with cognitive decline.

Filling in the survey after each session provided us with feedback on the evolution of participants’ perception of both usability and therapeutic benefits, as compared to the objective progress measured while performing the scheduled exercises. Surveys were also intended to contribute to detecting the system’s weaknesses and to identify future areas for improvement.

### 3.3. Physical Exercise Routines

Patients were asked to perform 20 shoulder exercises that covered flexion, abduction, and rotation joint movements and six knee exercises that covered retraction, extension, and protraction. Each user performed in each session a number of exercises depending on their physical condition. Participants were always guided by a professional physiotherapist.

Experiments focused on two particular exercises for shoulder and knee, respectively, which were carried out by all the participants in the six sessions. They were the shoulder abduction and the double leg squat. A standard session routine for both exercises consisted of three series, each of 10 to 15 repetitions with target angles established by the physiotherapist. The system monitored each repetition, controlled motion correction during the exercise performance, and measured the angle of the maximum range of motion.

In the case of shoulder abduction, the angle of the arm with respect to the body was measured in each repetition. The higher the angle, the better the performance. In the case of the double leg squat, the angle between the thigh and the lower leg was measured in each repetition. The correct range of motion is from 180º (upright position) to 90º (the thighs are parallel to the floor). Thus, unlike the shoulder exercise, the lower the angle, the better the performance. The average angle over all repetitions of an exercise performed by a participant in one session was considered the performance measure of that participant in that session.

### 3.4. Statistical Tests

Considering the presence of outliers and the noticeable differences in sample variances among sessions, genders and age ranges, the non-parametric Mann–Whitney U test was chosen to determine whether (or not) two series of results can be assumed to come from the same distribution (*H*_0_). In addition, to mitigate the impact of subject variability, the Wilcoxon signed-rank test was also applied to paired series. Test implementations were provided by the module scipy.stats from the Python library SciPy.

## 4. Experiments and Results

Results were examined from three different perspectives of analysis:Analysis of survey scoresAnalysis of physical achievementsCorrelations between survey scores and physical achievements.

Studies by age and by gender were carried out under the three perspectives in order to find out if any of these factors entail significant differences in the degree of acceptance or in the effectiveness of the system. This information could also determine how to adapt the tool to bring it closer to the expectations of people that fit a particular demographic profile.

### 4.1. Analysis of Survey Scores

The survey results were analyzed separately by age range (Figure 17) and gender (Figure 18). In both figures, score distributions by sessions were shown through boxplots. Boxes extend from the lower to upper quartile values, orange marks illustrate the median locations, whiskers are set at 10th and 90th percentiles, and extreme values are represented by points beyond the whiskers.

Figure 16 shows the evolution of score distributions over time for each considered age range. When focusing on the diagram corresponding to the range [65, 69], an increasing series of distribution medians (orange marks) is observed, which suggests a progressive approval of the tool by participants of these ages over time. In the other two ranges, the series of medians did not show a regular behavior in the early sessions, but both series ended up achieving high acceptance scores (4–5) from all the participants at sessions 5 and 6. The small variances of the distributions from sessions 5 and 6 in the three diagrams (represented by narrow boxes) mean that participants agreed to assign high acceptance scores (4–5) to all survey questions.

The observed differences were statistically evaluated by taking sessions 2, 4, and 6 as reference points. They were chosen to better discern trends in the evolution of the degree of acceptance of the system, thus avoiding the instability inherent in consecutive measurements in time series as well as potential misunderstandings of the first session. Table 4 summarizes the p-values obtained from contrasting the selected sessions using the Mann–Whitney U and the Wilcoxon tests. The differences were not relevant (for α = 0.05) between sessions 2 and 4, but they were very significant between sessions 4 and 6, where the margin for rejecting the null hypothesis (which establishes no difference) is clearly reduced with the increase of age. That is, until session 4, there were no significant changes in the scores assigned to the questions of the usability survey, which suggests initial doubts in the use of the system. By contrast, the highly significant increments in the evaluation of the system found in session 6 indicate that the participants finally understood and enjoyed the system.

Figure 18 illustrates the score distributions of questions by women (left) and men (right) in the six scheduled sessions over time. At first glance, the diagrams reveal differences between the acceptance scores given by women and men. The series of medians (orange marks) in the female view (left), which can be interpreted as a series of the average score that women gave to the system, shows a clearly increasing trend. The same series in the male view (right) shows a more irregular behavior. However, both series of medians eventually converged to the maximum score (5) in the sixth session.

Table 5 includes the p-values resulting from applying the two non-parametric statistical tests (Mann–Whitney U and Wilcoxon) in the assessment of differences between the score distributions of sessions 2 and 4 and of sessions 4 and 6. The Mann–Whitney U test identified significant gains (*p*-value < 0.05) in the female scores from session 2 to 4 and from session 4 to 6, while in the case of male scores, the only changes that were found to be significant were from session 4 to 6. The Wilcoxon test confirmed the meaningful changes between sessions 4 and 6. The fact that the Mann–Whitney U test found significant the two transitions studied (2–4, 4–6) in the series of female scores as opposed to only one transition (4–6) in the series of male scores suggests a more progressive acceptance of the system by women.

### 4.2. Analysis of Physical Achievements

As described above, each participant took part in six sessions over 15 days, performing two physical routines (shoulder abduction and double leg squat) in each session, with each routine consisting of 10 to 15 repetitions aimed at reaching a target angle. During the execution of each repetition, the system monitored whether it was being performed correctly and measured the angle of the maximum range of motion. The average angle over all repetitions was considered the participant’s physical achievement for the routine and session involved. The physical performance data collected in this study are publicly available at http://bit.ly/386bEDQ. Data are stored in a CSV file (Appendix A) which comprises the average angles computed for each participant, exercise and session.

Assuming accurate measurement techniques [25], the system will be considered effective or valid with respect to the goal of guiding older users in carrying out physical exercises if it is possible to statistically verify progress in the physical achievement of the participants over the sessions (over time), considering their average angles in the exercises and sessions involved.

Participants’ physical performances were also analyzed both in terms of age (Figure 19) and in terms of gender (Figure 20). Each boxplot represents the distribution of average angles computed for a physical routine in a particular session. Figure 19 depicts the evolution of physical achievements (average angles) over the sessions for each exercise: shoulder abduction in the top row and double leg squat in the bottom row. As explained in Section 3.3, the greater the angle in shoulder abduction, the better the performance, while in the case of the double leg squat, the lower the angle, the better the performance. A common behavior pattern is observed in both exercises, which can be summarized as follows: (1) there is a continuous progress in physical achievement for the three age ranges over the sessions; (2) absolute achievements decreases with age; (3) improvement in achievements (differences between two consecutive sessions) increases with age. The last point is particularly important, as it suggests that the benefits of using the system increases with age.

It is also remarkable that most distributions (boxplots) are very compact. This pattern, previously observed in [25], confirms the accuracy of the system in tracking and measuring movement, particularly in this context where older adults are supposed to have less regular and coordinated movements.

Two statistical analyses based on Mann–Whitney U and Wilcoxon signed-rank tests were conducted from the results summarized in Figure 19:For each combination of a given exercise and a given age range (i.e., each subplot), differences between the distributions (session 2, session 4), (session 2, session 6), and (session 4, session 6) were statistically validated both by the Mann–Whitney U test and by the Wilcoxon signed-rank test. All p-values resulting from applying both tests in all scenarios were comfortably less than α = 1 × 10^−2^, proving the significant improvements in the physical achievement over the sessions. Note that this holds for all ages in the two types of exercises.For each type of exercise, differences between each pair of analogous sessions in consecutive age groups were statistically assessed by the Mann–Whitney U test (here, the Wilcoxon signed-rank test does not apply). This was aimed at finding out if the system detects significant differences in the physical performances of two contiguous age groups in the same session. Although all p-values were much lower than α = 1 × 10^−3^ (strong verification of significant changes), the most relevant differences were found between the groups 65–69 and 70–74 with respect to differences between the groups 70–74 and 75–80. Beyond confirming the expected worsening of physical condition with age, these results provide clear evidence in favor of the value of the system as a domestic tool for assisting older adults of different ages in performing active aging routines.

Figure 20 shows the evolution of physical condition from a gender perspective. Differences were analyzed through the same statistical studies described above. Very significant differences were found (*p*-values < 1 × 10^−4^) in all comparisons involving even sessions (2, 4, and 6) within each subplot (one gender and one exercise), showing the continuous improvement in physical achievement also from the perspective of each gender. On the contrary, no significant differences were found between the results of women and men in any session of any exercise.

### 4.3. Correlations between Survey Scores and Physical Achievements

The last study assesses the relationship between the perceived benefit (survey scores) and the measured benefit (physical achievements). Given a subset of participants that meet some age or gender criteria and a particular exercise, the statistical dependence between the series of medians of their survey scores and the series of their average angles measured by the system over all sessions was evaluated by the Spearman’s rank correlation coefficient. For instance, in the case of the 23 people with ages in the range [65, 69], both series were made up of 138 elements (23 people × 6 sessions). Spearman’s method is a nonparametric measure of rank correlation between two variables, which does not assume anything about the distribution of the variables or about the nature of the relationship between them.

Table 6 and Table 7 show the Spearman’s rank correlation coefficients corresponding to data arrangements determined by age ranges and genders, respectively. Statistically significant correlations were obtained in all scenarios, although they were less relevant (higher *p*-values) in the gender analysis. This is an expected result, because people of the same gender involve all ages considered in the study, from 65 to 80 years. The weakest correlations determined by the largest *p*-values occurred for the older ages [75, 80] and for male participants. On the contrary, in women aged 65–69, a correlation of −0.70 (*p*-value = 7.0 × 10^−11^) was obtained from their results in the double leg squat, demonstrating a high level of coincidences between the questionnaire answers and the physical achievements. The minus sign in this exercise means that a smaller angle corresponds to a better performance. Previous results prove that older people, despite their poor technological training, can positively take advantage of multimedia tools aimed at fostering active aging if they experience visible benefits.

## 5. Conclusions

This work has presented an interactive system to assist older adults in performing physical exercises in domestic environments. The system combines the acquisition capabilities of a Kinect RGBD sensor with the communication potential of gamification and augmented reality. These interactive technologies have proved their effectiveness in the healthy aging context, as long as the resulting interactive elements are adapted to the skills and preferences of the elderly and provide useful feedback on the objective progress in their physical condition.

Experiments were intended to meet a twofold purpose: to measure the degree of acceptance of the system by older people and to assess the effectiveness of the system in measuring the physical achievement of older users when carrying out prescribed exercises. The acceptability and effectiveness of the system were measured in terms of a usability survey and statistically verified progress of physical achievement over a series of sessions with goals of increasing complexity. Three areas of analysis were considered: survey scores, physical achievements, and correlations between scores and physical achievements. Studies by age and gender were separately conducted. Results suggest several important conclusions:Statistical analysis of survey scores showed a progressive acceptance of the tool by older users.Statistical tests proved continuous improvement in the measurement of physical achievements in all age ranges and both genders in the two types of exercises. However, the greatest progress over the sessions was observed in the group of the oldest people (75–80).Correlation coefficients between perceived benefits, expressed through survey scores, and verifiable benefits, in terms of objective measurements of physical achievements, were obtained. Statistically significant correlations were found in all scenarios, although the most relevant were observed in the group of women aged [65, 69].

Summarizing, the proposed RGBD sensor-based interactive system has proved to be a valuable tool for promoting healthy aging activities, which allows measurable physical improvements in older users through attractive user interfaces adapted to the characteristics of the elderly.

Some future development lines will be focused on implementing more intuitive authentication methods based on biometric features (e.g., face, voice), interaction methods based on voice commands and adaptive training routines in which activities and objectives are automatically adjusted according to the user’s progression.

## Figures and Tables

**Figure 1 sensors-21-00417-f001:**
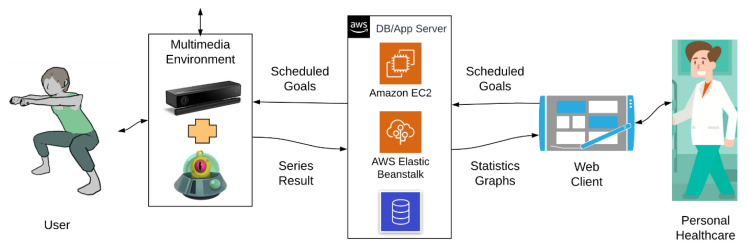
KineActiv system architecture [25].

**Figure 2 sensors-21-00417-f002:**
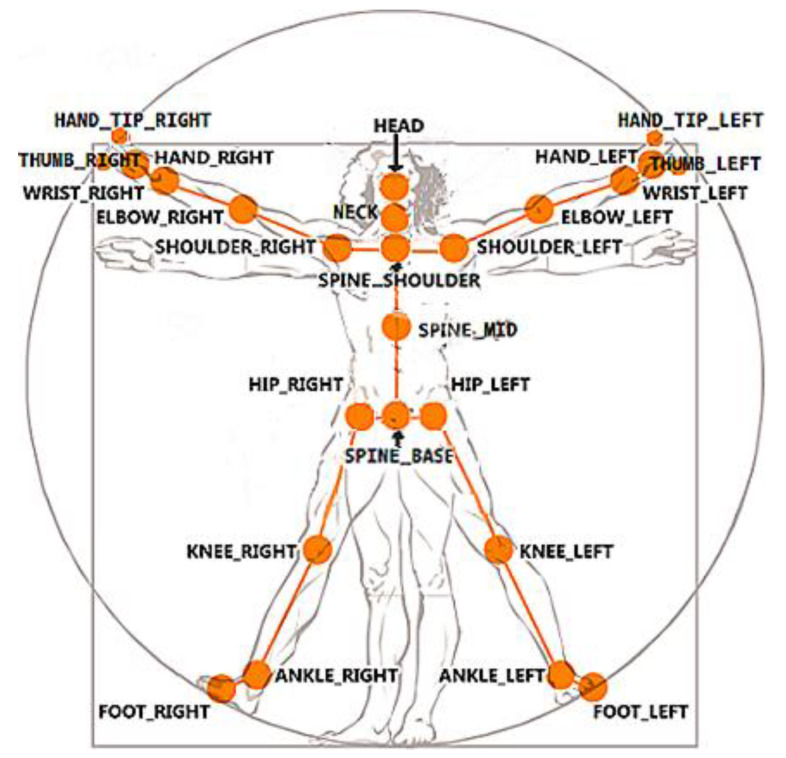
Skeletal model provided by the Software Development Kit (SDK) consisting of the 3D locations of 25 joints [37].

**Figure 3 sensors-21-00417-f003:**
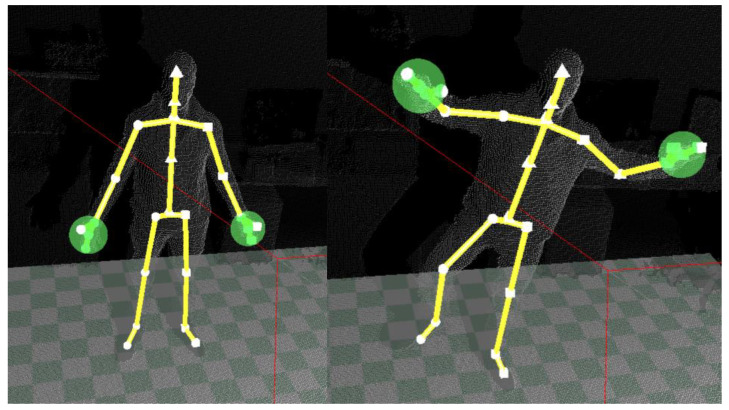
An example of body motion tracking through the built-in skeletal model.

**Figure 4 sensors-21-00417-f004:**
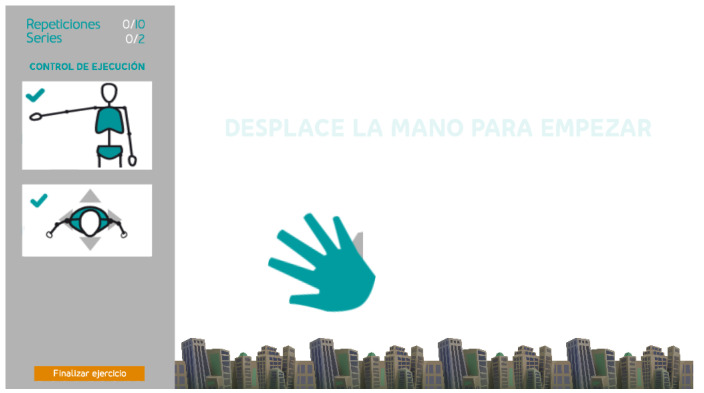
The system illustrates the action (a hand gesture) required to start the exercise.

**Figure 5 sensors-21-00417-f005:**
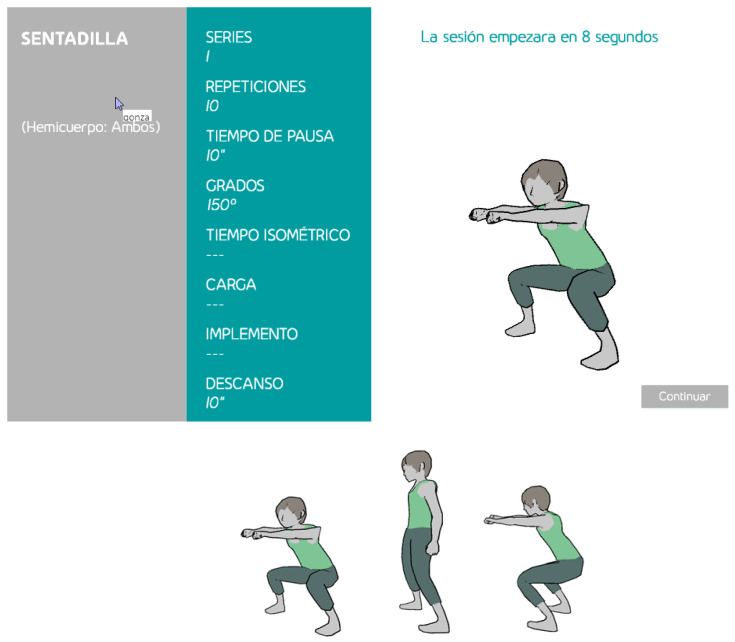
A 3D avatar illustrating how to perform the squat exercise. In the top, the screen with the scheduled exercise. In the bottom, some shots of the avatar motion sequence.

**Figure 6 sensors-21-00417-f006:**
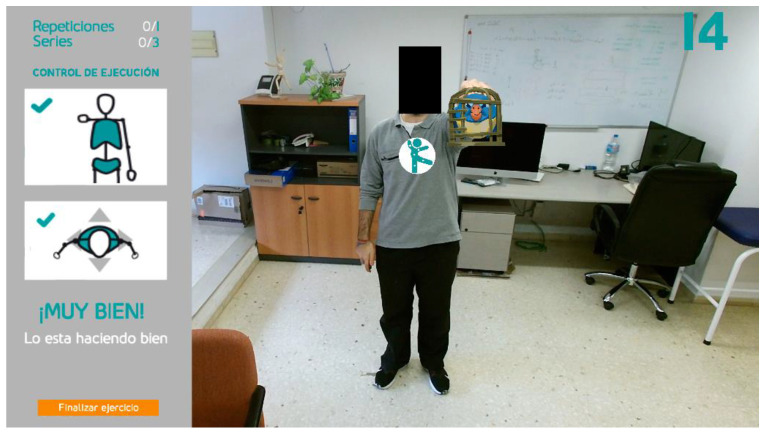
An Augmented Reality (AR) element embedded in a gamified environment: put the bird into the cage.

**Figure 7 sensors-21-00417-f007:**
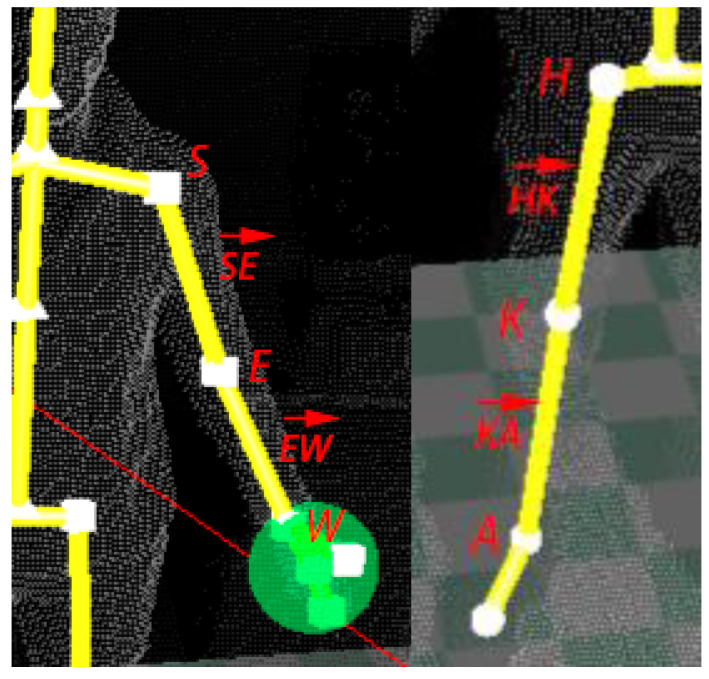
Joints and vectors involved in a calibration step for a shoulder exercise (**left**) and a knee exercise (**right**).

**Figure 8 sensors-21-00417-f008:**
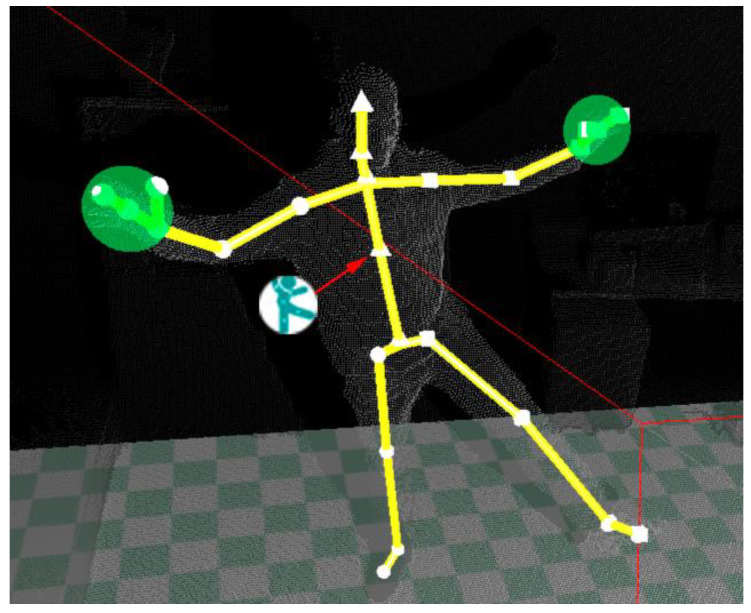
The mark (a circle enclosing the system logo) is placed on the position estimated for the SPINE_MID joint, which is located around the thoracic T5 vertebrae.

**Figure 9 sensors-21-00417-f009:**
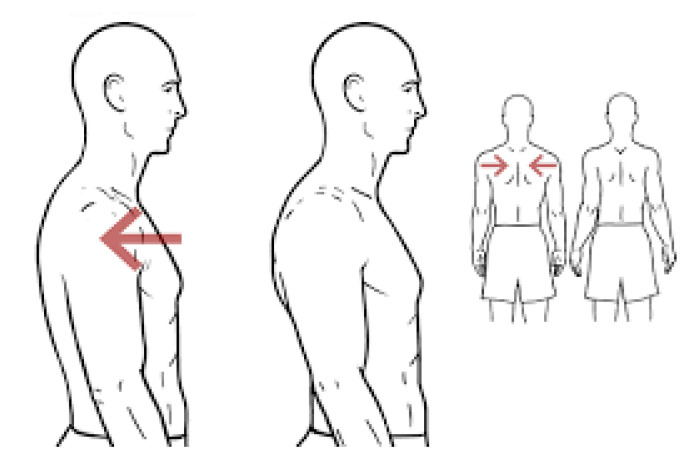
Scapular retraction (https://workoutsprograms.com/ejercicios/retraccion-escapular).

**Figure 10 sensors-21-00417-f010:**
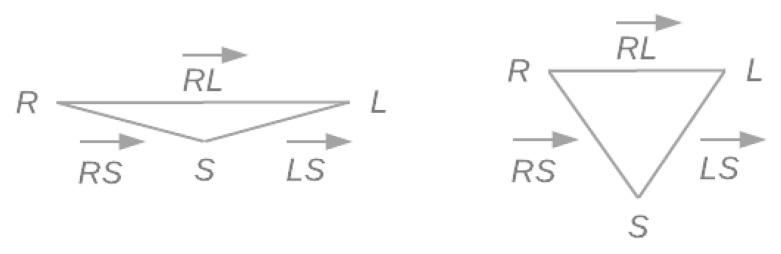
Triangle defined by the joints SHOULDER_RIGHT, SPINE_SHOULDER, and SHOUDER_LEFT (from a top view). On the left, a representation of the resting state of these joints. On the right, deformation caused by the correct execution of a scapular retraction.

**Figure 11 sensors-21-00417-f011:**
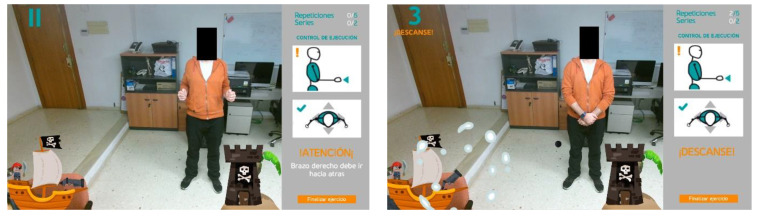
Gamification environment designed for isometric scapular retraction. On the left, while the user holds muscle tension, the cannon gains firepower. On the right, if the accumulated power is sufficient, the projected bullet (black ball in the center) will hit the tower.

**Figure 12 sensors-21-00417-f012:**
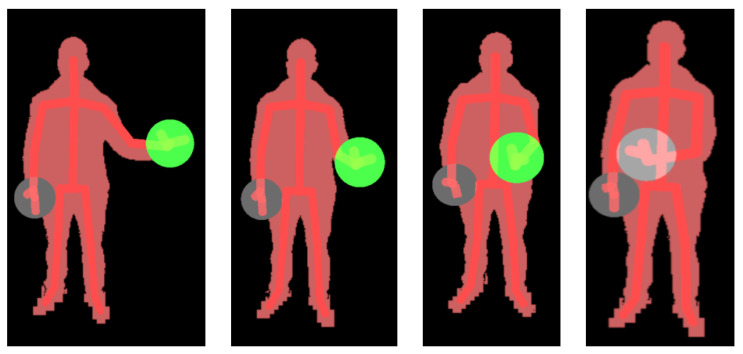
Gesture recognition: from left to right, the circle represents the tracking of the hand from one side to the other. The gray circle indicates that the final pose has been identified.

**Figure 13 sensors-21-00417-f013:**
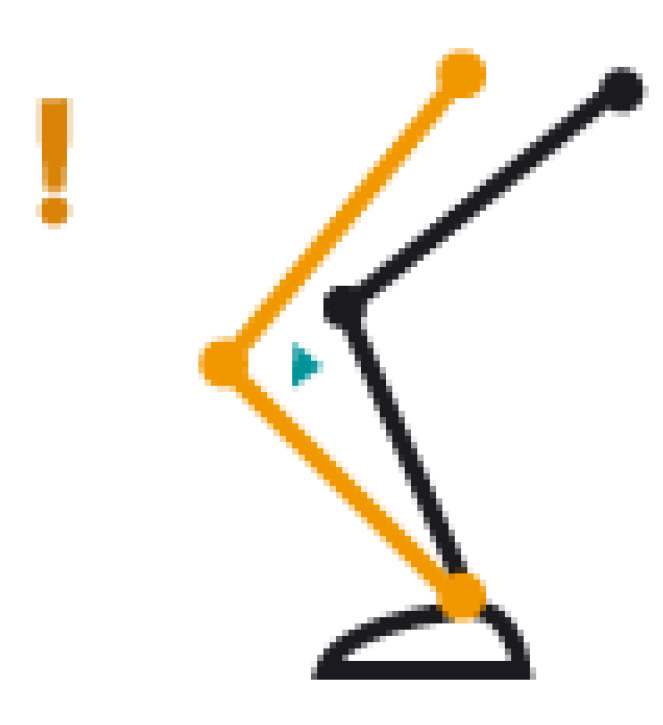
Correction of the position of the knees during the execution of the squat exercise.

**Figure 14 sensors-21-00417-f014:**
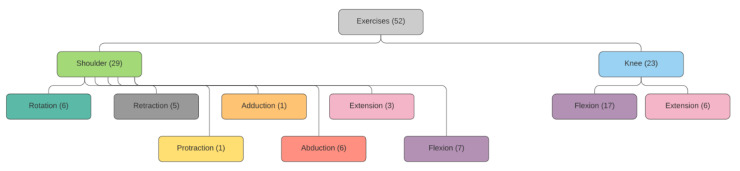
Set of implemented exercises.

**Figure 15 sensors-21-00417-f015:**
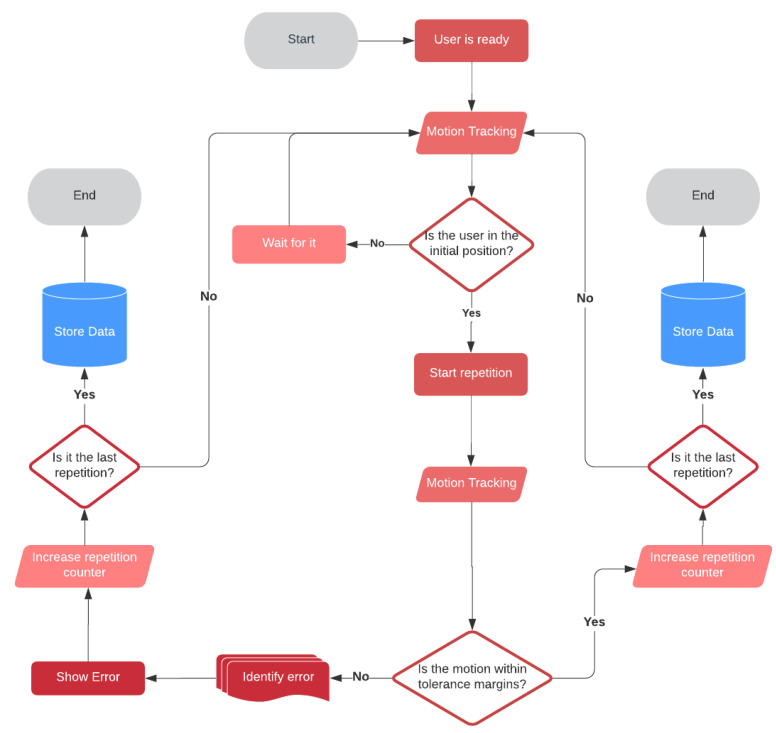
Exercise execution flow diagram.

**Figure 16 sensors-21-00417-f016:**
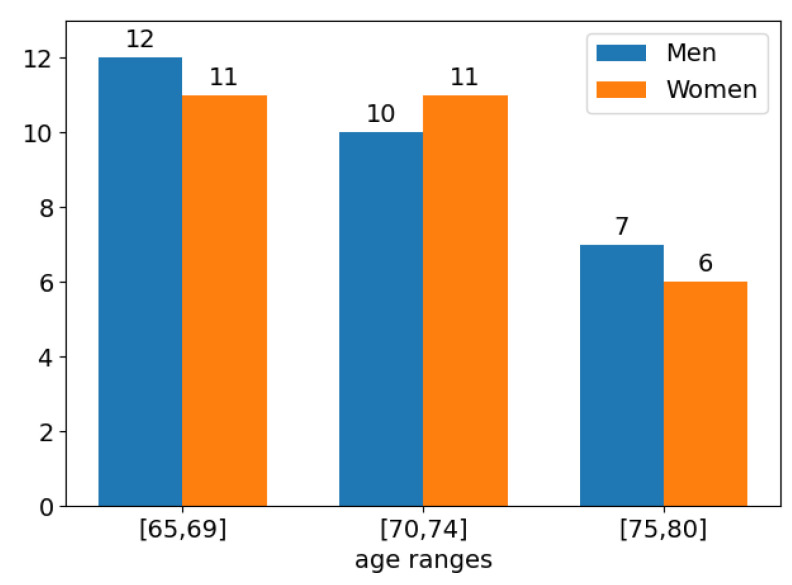
Distribution of participants by age range.

**Figure 17 sensors-21-00417-f017:**
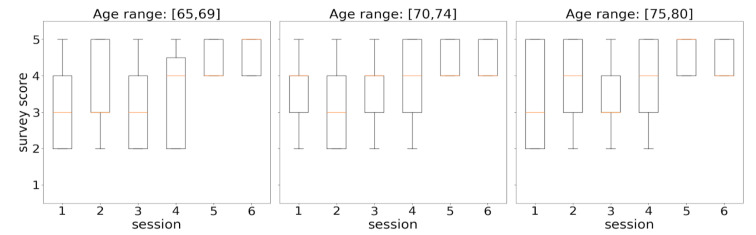
Distributions of survey responses by session from an age perspective.

**Figure 18 sensors-21-00417-f018:**
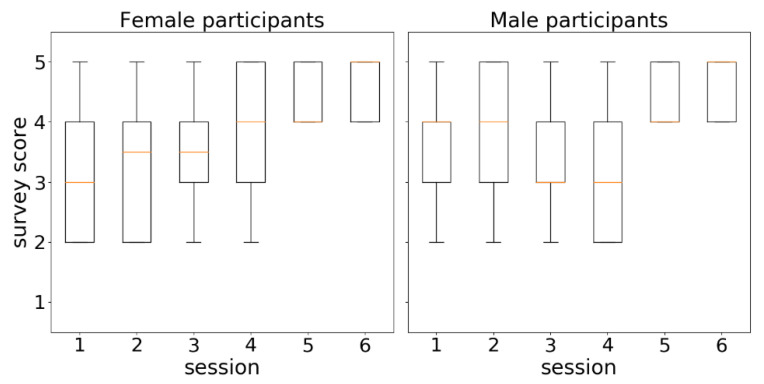
Distributions of survey responses by session from a gender perspective.

**Figure 19 sensors-21-00417-f019:**
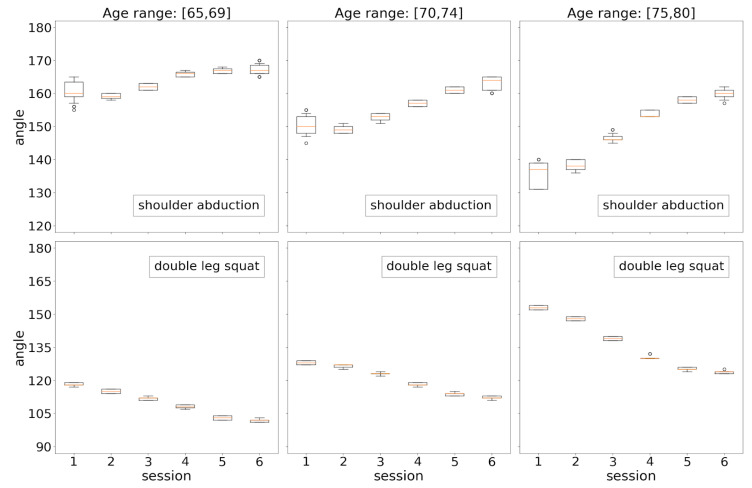
Distributions of exercise results by session from an age perspective.

**Figure 20 sensors-21-00417-f020:**
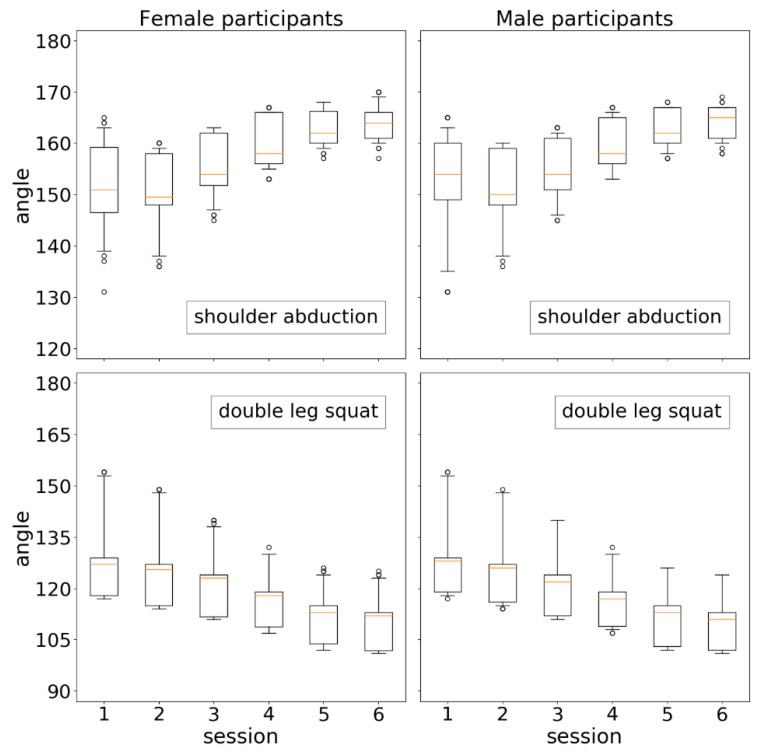
Distributions of exercise results by session from a gender perspective.

**Table 1 sensors-21-00417-t001:** Comparison of recent publications on active aging and this work, considering both features of the systems and methods for evaluating their effectiveness.

Related Works (Active Aging)	1 [31]	2 [32]	3 [33]	4 [34]	5 [27]	6 [29]	7 [30]	8 [28]	OURS
Year of publication	2020	2020	2020	2020	2018	2018	2020	2019	2020
Kinect-based	X	X	X	X	X	X	X	X	X
AR environment	-	-	X	-	X	-	-	X	X
Gamification	-	-	X	-	X	-	-	X	X
Gesture-based interactions	-	-	-	-	X	-	X	-	X
Usability study	-	-	-	-	-	-	-	-	X
Exercises for upper limbs	-	-	X	X	X	X	-	X	X
Exercises for lower limbs	-	-	-	X	X	X	-	X	X
Real-time motion control	-	-	-	-	-	-	-	-	X
Real-time measurement	X	X	-	X	-	-	-	-	X
Statistical assessment	-	-	-	-	-	-	-	-	X
Participants by gender: F/M	80	6	3	10/10	46	442	1/3	52	28/29
Participants ages (average age or age range)	-	23,3	[43, 62]	[70, 90]	69,3	73,3	[60, 75]	[60, 90]	[65, 80]
In-home assistance	-	-	X	-	-	-	-	-	X

**Table 2 sensors-21-00417-t002:** The System Usability Scale (SUS) models.

1	I think that I would like to use this system frequently
2	I found the system unnecessarily complex
3	I thought the system was easy to use
4	I think that I would need the support of a technical person to be able to use this system
5	I found the various functions in this system were well integrated
6	I thought there was too much inconsistency in this system
7	I would imagine that most people would learn to use this system very quickly
8	I found the system very cumbersome to use
9	I felt very confident using the system
10	I needed to learn a lot of things before I could get going with this system

**Table 3 sensors-21-00417-t003:** Questionnaire designed for the acceptability study. Each question is classified into usability categories and related with SUS items.

Question	Category	SUS Coverage
I would use the system again	Satisfaction	1
I have correctly understood the messages while doing the exercise	Ease of use	2,3,7,8
The images have been sufficiently explanatory and have helped me correct my posture.	Ease of use	2,3,7,8
The rules of the games are clear and understandable	Ease of use	2,3,7,8
It was easy for me to follow the instructions of the exercises	Ease of use	2,3,7,8,10
In future occasions I could use the system without the supervision of a physical therapist	Ease of use	2,3,4,7,8,10
I have felt very good, and I have enjoyed doing the exercises	Happiness	9
The use of games in rehabilitation is motivating for me	Importance	-
These types of tools help my recovery	Usefulness	-

**Table 4 sensors-21-00417-t004:** Statistical tests (Mann–Whitney U/Wilcoxon) to assess changes in survey responses over time (sessions 2, 4, and 6). *p*-values in bold represent significant differences for α = 0.05.

*p*-Values	Session 2 vs. Session 4	Session 4 vs. Session 6
[65, 69]	0.96/0.99	3.2 × 10^−22^/3.2 × 10^−21^
[70, 74]	0.08/0.12	6.8 × 10^−16^/2.0 × 10^−16^
[75, 80]	0.20/0.25	7.4 × 10^−10^/2.9 × 10^−10^

**Table 5 sensors-21-00417-t005:** Statistical tests (Mann–Whitney U/Wilcoxon) to assess changes in survey responses over time (sessions 2, 4, 6). *p*-values in bold represent significant differences for α = 0.05.

*p*-Values	Session 2 vs. Session 4	Session 4 vs. Session 6
Females	0.042/0.054	7.9 × 10^−17^/1.3 × 10^−19^
Males	0.166/0.183	1.8 × 10^−30^/9.7 × 10^−27^

**Table 6 sensors-21-00417-t006:** Spearman’s rank correlation coefficients (*p*-values) to assess statistical dependence between the rankings of survey responses and physical achievements by age range.

	[65, 69]	[70, 74]	[75, 80]
Shoulder abduction	0.50 (6.3 × 10^−10^)	0.56 (1.4 × 10^−11^)	0.43 (7.3 × 10^−5^)
Double leg squat	−0.57 (4.3 × 10^−13^)	−0.55 (4.1 × 10^−11^)	−0.48 (9.4 × 10^−6^)

**Table 7 sensors-21-00417-t007:** Spearman’s rank correlation coefficients (*p*-values) to assess statistical dependence between the rankings of survey responses and physical achievements by gender.

	Female	Male
Shoulder abduction	0.37 (9.2 × 10^−7^)	0.35 (3.0 × 10^−6^)
Double leg squat	−0.34 (5.1 × 10^−6^)	−0.27 (2.6 × 10^−4^)

## Data Availability

The physical performance data collected in this study are publicly available at http://bit.ly/386bEDQ.

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
