# Peer review of "A Kinect-Based Interactive System for Home-Assisted Active Aging"

_sensors, 2021, doi:10.3390/s21020417_

Round 1

Reviewer 1 Report

This work presents a Kinect-based interactive system aiming to assist older adults in performing physical exercises. The system in this study, i) is based on a previous implementation on a Kinect sensor and ii) tries to combine gamification and augmented reality for a cohort of elderly people.

The technical details are not presented in depth, nor how the gamification and the AR is used (apart from the obvious functions Kinect has). It is important these details to be presented (especially since they authors claim a qualification and validation of usability). Also, for reasons of competence the authors need to present similar works (and show innovation and differences/ contribution of this study).  These sections of the paper need a lot of improvement. Also, implementation and evaluation methodology must be extended.

The validation experiments in this study were supposed to i) evaluate the degree of acceptance of the system by users and assess its validity (?) in promoting effective routines for improving their physical condition – (please explain if this is to prove that the system actually can deliver the expected output – i.e. work properly). It is this reviewer understanding that this system is based on a previously existing system (studied and created and now adapted for elderly). In that respect it is important to clarify the outcome and actual benefit for using it. The authors claim that this is “… a valuable tool for healthy aging activities, which allows measurable physical improvements in older users through attractive user interfaces adapted to the characteristics of the elderly”. But the presented results are far to even closely justify this  - this section must be improved in order for the paper to be complete and solid.

In particular,

  • The usability study is not presented (how the system was thus accepted?)
  • The survey scores analysis concludes into the following results (if not – please explain/specify/analyze/extent if possible) as quoted from the paper.
  • “These results suggest that initial doubts in understanding the tool had been resolved in the final sessions”
  • “That is, results suggest a better adaptation of women to the use of the tool”

These two points actually need to be further extended and reach a conclusion towards what the authors need to conclude to!

  • Physical results analysis:

Explain: average angles resulting from the execution of a given physical routine in a particular session by participants who meet age or gender How the following argument can be derived from what is presented? : “improvement in physical condition over the sessions for each exercise” – provide facts and reasons for this result deduction

This paper needs major revision and resubmission

Author Response

The technical details are not presented in depth, nor how the gamification and the AR is used (apart from the obvious functions Kinect has). It is important these details to be presented (especially since they authors claim a qualification and validation of usability).

We appreciate the Reviewer's suggestions.

In order to address the Reviewer's comment regarding how the gamification and the AR are used, Sect. 2.4 (Gamified environment with rewarding policies) has been extended to include a second example of gamification environment: a pirate ship fires a cannonball with a force proportional to the time the muscle tension is held during an isometric exercise. This is illustrated in two new snapshots of the exercise while the user is interacting with the game, along with an explanation of its rationale. This game was designed to make isometric exercises such as scapular retraction more fun. The algorithm to control the correct execution of this exercise is previously explained in that same section.

Let us take the opportunity to recall that the original manuscript, in section 2.3 (AR-based user interface), already presented and explained a first game (Fig. 6). Both games are part of the 34 AR-based gamified environments designed to keep up users’ motivation, as indicated in the new section 1.2 referring to the contributions of this work.

The beginning of section 2.4 has also been extended to describe the criteria used to design the games and how they have been assigned to the different types of exercises, in order to provide consistency to the user interaction.

In addition, new details about the logic that controls a typical exercise have been added in Sec. 2.7 through a flow diagram and supplementary explanations.

Also, for reasons of competence the authors need to present similar works (and show innovation and differences/ contribution of this study).

We agree with the Reviewer's opinion that this issue was insufficiently addressed in the original manuscript. As part of the redesign of Section 1 (Introduction), a new subsection (1.1) has been created to extend the discussion of recent works on Kinect-based systems related to active aging with the objective of positioning this work in the field of active aging systems based on Kinect sensors. This subsection also includes a new table (Tab. 1) that summarizes a comprehensive comparison between the most relevant works found and our system, regarding a number of common criteria that encompass features of the proposed systems and methods for evaluating their effectiveness. To our knowledge, no other work has proposed a system for home-assisted use specifically aimed at active aging, capable of accurately controlling and measuring movement during the execution of proposed exercises. Furthermore, we have not found any studies that make a double evaluation of the system in terms of acceptability and validity.

Also, implementation and evaluation methodology must be extended.

We appreciate the Reviewer's suggestions.

Section 2.7 (Implementation) has been extended by adding a general flow diagram that illustrates the general logic of supervision and control of any exercise. An explanatory text has also been added to help understand the process represented in the diagram.

Section 3 (Experimental methodology) has been completely revised. Many pieces of text have been modified in order to provide clearer explanations of the methods involved and justify the decisions made. In particular, Section 3.2 has been completely rewritten to introduce the fundamentals of the designed survey, as well as its relationship to well-known standards.

The validation experiments in this study were supposed to i) evaluate the degree of acceptance of the system by users and assess its validity (?) in promoting effective routines for improving their physical condition – (please explain if this is to prove that the system actually can deliver the expected output – i.e. work properly). [...] In that respect it is important to clarify the outcome and actual benefit for using it. The authors claim that this is “… a valuable tool for healthy aging activities, which allows measurable physical improvements in older users through attractive user interfaces adapted to the characteristics of the elderly”. But the presented results are far to even closely justify this  - this section must be improved in order for the paper to be complete and solid.

We understand the Reviewer's doubts.

Although we believe that both results, the usability survey scores and measurements from physical achievements, provide enough evidence on the effectiveness of the system, after the reviewer’s comments, we realize that we were not able to explain them properly; important ideas remained inaccurately or insufficiently exposed.

The key criterion to establish that expectations regarding acceptability or effectiveness were met were the presence of statistically significant improvements between measurements (survey responses, physical achievements) spread over a period of 15 days.

In this second version of the manuscript a very important effort has been made to extend and rewrite more clearly and precisely the presentation and discussion of results. Particular emphasis has been placed on explaining experiment decisions. More precisely, we have rewritten large parts of sections 4.1 and 4.2.

From the end-user perspective, we consider that the system helps older users to correctly perform age-appropriate physical exercises, and to physiotherapists to remotely prescribe and control personalized exercises.

The usability study is not presented (how the system was thus accepted?) The survey scores analysis concludes into the following results (if not – please explain/specify/analyze/extent if possible) as quoted from the paper: 1) “These results suggest that initial doubts in understanding the tool had been resolved in the final sessions”; 2) That is, results suggest a better adaptation of women to the use of the tool”.

We appreciate the Reviewer's suggestions.

Important changes have been made to better explain and justify the design of the usability questionnaire, and to present the discussion of results more clearly.

Section 3.2 has been completely rewritten to introduce the fundamentals of the designed survey, as well as its relationship to well-known standards. These foundations will help to explain the suitability of the proposed model to collect information from older people.

The conclusions highlighted by the Reviewer (initial doubts and better adaptation of women) were drawn from results shown in Figures 17 and 18 and in Tables 4 y 5. New and more detailed explanations have been added to section 4.1 in order to better support both conclusions.  Changes are highlighted in red fonts.

Physical results analysis. Explain: average angles resulting from the execution of a given physical routine in a particular session by participants who meet age or gender

The question raised by the Reviewer is entirely appropriate.

We now realize that the phrase the Reviewer refers to was intended to summarize a lot of information, and the result was very cryptic. That sentence has been replaced by two paragraphs with a detailed explanation (the first two paragraphs of section 4.2).

How the following argument can be derived from what is presented? : “improvement in physical condition over the sessions for each exercise” – provide facts and reasons for this result deduction.

The Reviewer's comment has allowed us to identify incorrectly used terms and poorly explained results. The following changes are derived from this comment.

Some terms such as "health condition" or "physical condition" are now avoided as they are too unspecific, could create unreasonable expectations and could mislead the reader. Instead we now use "physical capacities required for a particular exercise" or “physical achievements”.

In this regard, we have assumed that achieving a greater average range of motion (physical achievement) over multiple repetitions of an exercise with respect to previous sessions, can be explained by an improvement in the physical capacities required for that exercise.

We believe that this change in terminology, together with the new arguments and explanations, should contribute to clarify and better support the conclusions proposed from the experiments.

Reviewer 2 Report

The KineActiv  system presented by the authors one year ago in the same journal is extended in this paper that is generally speaking well written and comprehensive. Its aim is to replace the supervision of a physiotherapist and give elderly people the opportunity to exercise by themselves and assess their progress using augmented reality facilities.

Undoubtedly, the work presented by the authors is important to a significant problem faced by the aging population. My main concerns are the following:

1) it is not well clarified what is the additional contribution of this paper compared to [24]. I had a quick look at [24], there are certainly more details presented in the last paper but the differences between the two version have to be clarified in the present version.

a) Are the bullets in lines 100-108 the ones that differenciate the new paper? If the survey presented is the main contribution of the new paper, then it seems that it is not enough as a new contribution.

b) Are there new AR facilities that were not supported in [24]?

c) Is the previous version extended to monitor other body parts than upper limb as I think it was the case in [24]? it has to be clarified

2) there are no comparable results with other approaches in the literature.

a) Even if there are results that there are not directly comparable, a table that would summarize the different approaches and what is measured in each approach would be useful

b) the authors should find objective metrics to compare. For example how accurately are the body part positions recognized after calibration? Although even in this case Kenect would be credited with this accuracy because I do not think that the authors have contributed much to this point (of precise body part position measurement)

Author Response

1) it is not well clarified what is the additional contribution of this paper compared to [24]. I had a quick look at [24], there are certainly more details presented in the last paper but the differences between the two versions have to be clarified in the present version.

a) Are the bullets in lines 100-108 the ones that differentiate the new paper? If the survey presented is the main contribution of the new paper, then it seems that it is not enough as a new contribution.

b) Are there new AR facilities that were not supported in [24]?

c) Is the previous version extended to monitor other body parts than upper limb as I think it was the case in [24]? it has to be clarified

We agree with the Reviewer that presentation of contributions and discussion of the state of the art was insufficiently addressed in the original manuscript. All doubts raised by the reviewer are therefore entirely appropriate.

A new subsection (1.2) specifically dedicated to presenting and explaining in detail the contributions of this work has been added to the revised version. A first part points out the most important differences between the new system and KineActiv (the previous system on which it is inspired). A second part highlights the contributions regarding the current state of the art on Kinect-based systems designed to promote physical exercise among older people.

We believe that the new section clearly establishes the novelty and value of the current proposal.

2) there are no comparable results with other approaches in the literature.

a) Even if there are results that there are not directly comparable, a table that would summarize the different approaches and what is measured in each approach would be useful

b) the authors should find objective metrics to compare. For example how accurately are the body part positions recognized after calibration? Although even in this case Kenect would be credited with this accuracy because I do not think that the authors have contributed much to this point (of precise body part position measurement)

We agree with the Reviewer that the original manuscript does not cover in depth the exploration of the related literature.

To address this deficiency, a new subsection (1.1) has been created to extend the discussion of recent works on Kinect-based systems related to active aging.

Following the Reviewer's suggestion, this subsection also includes a new table (Tab. 1) that summarizes a comprehensive comparison between the most relevant works found and our system, regarding a number of common criteria that encompass features of the proposed systems and methods for evaluating their effectiveness.

From the reviewed works and the analysis of Tab. 1, we can conclude that, to the best of our knowledge, there is no proposal directly comparable to ours in terms of functional scope, control algorithms, and diversity of user environments and interaction modes.

Regarding item b), on the need to measure the precision of the system, this is a point discussed in the article in which KineActiv (previous proposal) was introduced (Sensors, 2019), through a section titled "System Accuracy". In that section, the system measurements and measurements made manually using a goniometer during a series of exercises performed by a professional physiotherapist were compared. Relative errors were provided as precision measures. In two new paragraphs added at the beginning of the section "4.2 Analysis of physical achievements" in the revised manuscript, this study is referred to as a premise to evaluate the effectiveness of the system through the following sentence "Assuming accurate measurement techniques [25], the system will be considered effective...".

In addition, we accept as additional evidence of the precision of the system the existence of compact distributions, with very little deviation with respect to their medians, obtained for the same age group in the same session (Fig. 19). Summarizing, we believe that the consistency of the measurements over the sessions is also an objective evidence of the system's precision. This reasoning is considered in Sec. 4.2 through the following paragraph:

It is also remarkable that most distributions (boxplots) are very compact. This pattern, previously observed in [25], confirms the accuracy of the system in tracking and measuring movement, particularly in this context where older adults are supposed to have less regular and coordinated movements.

Round 2

Reviewer 1 Report

the authors tried to provide a improved revision of the paper. 

their aim was to support answer all of the comments. 

Still though the paper lucks of a concrete and stable methodological approach towards supporting the results, and more than that it is not obvious how it differentiates from no 25 ref. 

Thus, the paper focus and purpose must be restated. 

What do the authors aim to? show the acceptability of a system in terms of usage? This must be accompanied by related technologies for UI etc. etc. and respected experiments in order to allow the participants of the study to provide solid results ... (i.e. how log was the study? where they using a different system? did they manage to use it for so long in order to improve their wellbeing?) 

Also some results makes no sense. There is no justification why to try and emphasise on the difference of acceptance between men/women??

The analysis on physical achievements is rather poor. the results show that people followed instructions by the system and the system managed to guide them - but this is subjective  - In order to make the argumentation we need objective/repetitive results and it seems from the experiment design these do not exist (please elaborate if they are) . 

Thus the paper needs improvement for the study (the authors extended the section on the degree of acceptance by older people) - and clearly show Also how it was objectively accessed in respect of its validity  in improving physical capacities in the elderly 

Author Response

COVER LETTER

PAPER TITLE: A Kinect-based interactive system for home-assisted active aging

MANUSCRIPT ID: sensors-1038325

The authors would like to thank the Reviewers for their valuable help in improving the quality of this work.

After the first round of reviews, the Reviewer 2 has suggested to accept this paper. Thank you.

In contrast, Reviewer 1 still questions major issues of the manuscript. Each one is analyzed and discussed separately.

New changes in the manuscript can be found in red fonts (text added or modified) or in strikethrough format (text to be removed).

REVIEWER 1

the authors tried to provide a improved revision of the paper. their aim was to support answer all of the comments. Still though the paper lucks of a concrete and stable methodological approach towards supporting the results, and more than that it is not obvious how it differentiates from no 25 ref.

We regret that the many modifications made to the original version of the manuscript during the first round did not seem sufficient to Reviewer 1.

Among the most important changes we can highlight the two new sections dedicated to related works (1.1) and to contributions (1.2), respectively. We believe that the novelty of the work was explicitly established in both sections and, in particular, the differences with respect to the previous work [25]. Five areas where the new system differs significantly from the previous proposal were identified, listed and discussed in the Sect. 1.2.

Thus, the paper focus and purpose must be restated. What do the authors aim to? show the acceptability of a system in terms of usage? This must be accompanied by related technologies for UI etc. etc. and respected experiments in order to allow the participants of the study to provide solid results ... (i.e. how log was the study? where they using a different system? did they manage to use it for so long in order to improve their wellbeing?)

The last observation of the Reviewer has made us think of the possibility we have not achieved to clarify what the specific focus of the paper is.

As explained in the manuscript, the experiments involved 57 elderly participants for 15 days in a center specialized in elderly care, during which 6 physical sessions were scheduled with at least 2 common exercises for shoulder and knee, respectively (see Sect. 3).

That is, experiments were conducted in a test environment in which the participants performed exercises of increasing difficulty determined by increasingly challenging target angles.

In our opinion, from the analysis of results, we understand that the following points were objectively verified:

  • The system was able to quantitatively measure significant progress (supported by two nonparametric statistical tests) in the physical achievement of the participants (not in their physical abilities), guided by a series of goals of increasing complexity.
  • The system was sensitive to age, as significant differences were found between the three consecutive age ranges studied.

Furthermore, an improvement in the physical capacities of the participants was observed during the experimental sessions monitoring. However, strictly speaking, this can only be considered an observed (not measured) result.

In summary, we believe that the system has proven its ability to objectively measure progress in physical achievements attained by older people, although we do not have medium or long-term results that prove the improvement in the physical condition of the participants.

Following the Reviewer's suggestions, and in line with the previous analysis, the paper focus has been modified and clarified accordingly. The manuscript has been exhaustively revised in order to consistently reestablish objectives, analysis of physical results and conclusions. We hope that this new approach better fits the Reviewer's point of view.

With regard to the system acceptability, it was assessed in terms of a usability survey that was designed considering some commonly standard categories: Satisfaction, Ease of use, Happiness, Importance and Usefulness. Although we are aware of the limitations of the template, we also found it useful to capture user feedback. This information will be used to improve both the user interface and the survey itself for future studies.

Also some results makes no sense. There is no justification why to try and emphasise on the difference of acceptance between men/women??

It was not our intention to emphasize the differences between the results of women and men. Furthermore, we did not try to explain or judge them.

We used gender as any variable of the experimental study, just as we used the different age ranges. Our analysis was limited to a mere description of the observed results, in order to try finding variables in the experimental data that showed some statistical difference.

Most of the analysis carried out did not show statistically relevant differences between the behavior of women and men, except in a particular scenario of the survey responses (see Sect. 4.1, Tab. 5) and in the correlation study between the perceived benefit (survey scores) and the measured benefit (physical achievements) (see Sect. 4.3, Tab. 7).

In any case, we would be willing to correct or remove any reference that the Reviewer considers unnecessary or inappropriate.

The analysis on physical achievements is rather poor. the results show that people followed instructions by the system and the system managed to guide them - but this is subjective  - In order to make the argumentation we need objective/repetitive results and it seems from the experiment design these do not exist (please elaborate if they are) . Thus the paper needs improvement for the study (the authors extended the section on the degree of acceptance by older people) - and clearly show Also how it was objectively accessed in respect of its validity  in improving physical capacities in the elderly

As we have mentioned before, the experimental scope of this work is a pilot study with 57 older participants who underwent 6 physical sessions distributed over 15 days. Each session consisted of at least 2 exercises with between 10 and 15 repetitions in each case. The initial goal of this experimental study was to allow us to empirically prove the tool's ability to measure progress in physical achievement, and in our opinion the results and their analysis confirm it.

In the medium and long term, as part of a technology transfer action, a study on a much larger scale will be organized in which the objective will be to physiologically evaluate the improvement of the physical condition and well-being of the elderly. However, these goals are beyond the scope of the presented work.

Reviewer 2 Report

The initial manuscript was significantly extended by the users especially in the introduction explaining in more details the referenced approaches and where the proposed approach differs. New Table 1 is very helpful.

The authros also give much more details about how their system operates and how the survey was carried out.

In my comment concerning the objective metrics that have to be used for comparison answered that some precision metrics have been published in their previous work.

Although I insist that a survey about how their system helps old people is not adequately scientific sound for a technical paper like this, I think that the features of the proposed work are useful to be published. For this reason I suggest to accept this paper.

Author Response

The initial manuscript was significantly extended by the users especially in the introduction explaining in more details the referenced approaches and where the proposed approach differs. New Table 1 is very helpful.

The authors also give much more details about how their system operates and how the survey was carried out.

In my comment concerning the objective metrics that have to be used for comparison answered that some precision metrics have been published in their previous work.

Although I insist that a survey about how their system helps old people is not adequately scientific sound for a technical paper like this, I think that the features of the proposed work are useful to be published. For this reason I suggest to accept this paper.

We would like to thank the Reviewer for suggesting the acceptance of the manuscript.

The manuscript has been significantly improved thanks to the constructive comments made by Reviewers in the first round.
